# Past and Projected Weather Pattern Persistence with Associated Multi-Hazards in the British Isles

**Paolo De Luca** [1,*], **Colin Harpham** [2], **Robert L. Wilby** [1], **John K. Hillier** [1], **Christian L. E. Franzke** [3] **and Gregor C. Leckebusch** [4]

1. Geography and Environment, Loughborough University, Loughborough LE11 3TU, UK; R.L.Wilby@lboro.ac.uk (R.L.W.); J.Hillier@lboro.ac.uk (J.K.H.)
2. Climatic Research Unit (CRU), School of Environmental Sciences, University of East Anglia, Norwich NR4 7TJ, UK; C.Harpham@uea.ac.uk
3. Meteorological Institute and Center for Earth System Research and Sustainability (CEN), University of Hamburg, 20146 Hamburg, Germany; christian.franzke@uni-hamburg.de
4. School of Geography Earth and Environmental Sciences, University of Birmingham, Birmingham B15 2TT, UK; G.C.Leckebusch@bham.ac.uk
* Correspondence: p.deluca@lboro.ac.uk

**Abstract:** Hazards such as heatwaves, droughts and floods are often associated with persistent weather patterns. Atmosphere-Ocean General Circulation Models (AOGCMs) are important tools for evaluating projected changes in extreme weather. Here, we demonstrate that 2-day weather pattern persistence, derived from the Lamb Weather Types (LWTs) objective scheme, is a useful concept for both investigating climate risks from multi-hazard events as well as for assessing AOGCM realism. This study evaluates the ability of a Coupled Model Intercomparison Project Phase 5 (CMIP5) multi-model sub-ensemble of 10 AOGCMs at reproducing seasonal LWTs persistence and frequencies over the British Isles (BI). Changes in persistence are investigated under two Representative Concentration Pathways (RCP8.5 and RCP4.5) up to 2100. The ensemble broadly replicates historical LWTs persistence observed in reanalyses (1971–2000). Future persistence and frequency of summer anticyclonic LWT are found to increase, implying heightened risk of drought and heatwaves. On the other hand, the cyclonic LWT decreases in autumn suggesting reduced likelihood of flooding and severe gales. During winter, AOGCMs point to increased risk of concurrent fluvial flooding-wind hazards by 2100, however, they also tend to over-estimate such risks when compared to reanalyses. In summer, the strength of the nocturnal Urban Heat Island (UHI) of London could intensify, enhancing the likelihood of combined heatwave-poor air quality events. Further research is needed to explore other multi-hazards in relation to changing weather pattern persistence and how best to communicate such threats to vulnerable communities.

**Keywords:** weather patterns; LWTs; persistence; multi-hazards; urban heat island; CMIP5; RCPs

## 1. Introduction

Persistent weather patterns can translate into hazards such as heatwaves, poor air quality, drought, wildfires and episodes of flooding [1–4], with significant socio-economic losses [5,6]. Examples of such impactful episodes include the 2003 and 2010 European summer heatwaves that led to more than 100,000 deaths, reduced gross primary productivity of crops and, in the latter episode over Russia, about US $15 billion economic losses [7–10]. Similarly, summer 2013 in eastern China, was the hottest ever recorded in that region, with persistent and widespread heatwaves and droughts causing severe socio-economic impacts amounting to 59 billion RMB in losses [11]. Conversely, the extremely wet and

stormy 2013/14 winter over the United Kingdom (UK) was characterised by the passage of numerous low-pressure systems causing extensive pluvial, fluvial, coastal and groundwater flooding along with severe gales [12–14].

Natural hazards pose a significant socio-economic threat, yet their spatio-temporal co-occurrence (termed herein multi-hazards) are not yet fully understood [15,16]. Multi-hazards/risks research has developed considerably over the last decade [17–21], such that the United Nations Sendai Framework for Disaster Risk Reduction (UNDRR) [22] has called for multi-hazard approaches to disaster risk reduction. Multi-hazards are also known as compound events [15,23]. Examples of multi-hazard studies include interactions between earthquakes and landslides [24], multi-basin fluvial flooding and extra-tropical cyclones [25], fluvial and coastal flooding [26–28], extreme wet and dry hydrological events [29–31], and compound cold-wet dynamical extremes over different continents [32]. Considering natural hazards as physical processes that can interact across both temporal and spatial scales is of interest to decision makers such as government agencies, local businesses, emergency management services and (re)-insurance companies. Natural hazards can compound in various ways (i.e., occur simultaneously, as cascades or cumulatively) over a sufficiently long time-frame [22], and therefore their combined socio-economic impacts can exceed what was originally planned for, putting societies and economies under stress [15].

Daily atmospheric pressure patterns for the British Isles (BI) have been categorised according to the system of Lamb Weather Types (LWTs) [33]. This classification was originally subjective, meaning that daily weather patterns were assigned manually after inspection of weather charts. A few years after the first subjective classification of LWTs [33], an objective method was developed to classify daily atmospheric circulation according to LWTs [34]. Eventually, both the subjective and objective approach were compared [35] and objective LWTs were subsequently derived from reanalysis products [36]. The main novelty of the objective classification scheme is that it uses grid-point daily mean sea-level-pressure (SLP) analysis for a fixed observation time (such as 00:00 or 12:00 UTC) [37].

Previous studies have investigated links between weather patterns (or large-scale atmospheric circulation) and local extreme events, such as heavy rainfall, storms, floods and heatwaves [25,38–46]. The conventional approach to fluvial flooding analysis at the *single* catchment scale is being extended to frameworks with inter-related hazards, driven by global climate modes, covering multiple catchments [39]. Others show that the bias in simulating regional extreme precipitation days by an Atmosphere-Ocean General Circulation Model (AOGCM) is reduced by applying atmospheric circulation indices [41]. Moreover, weather patterns extracted from AOGCMs have also been used to downscale local climate variables, such as temperature, precipitation, radiation and humidity at local scales [43,47,48]. However, AOGCMs vary in their ability to simulate the frequency, seasonality and persistence of weather patterns at regional scales [42,43].

Some studies have linked heavy precipitation events to atmospheric circulation states, such as the 850hPa geopotential height field or integrated vapour transport (IVT) [40], and found connections between LWTs [33–35], and multi-basin fluvial flooding driven by extra-tropical cyclones (ETCs) [25]. In the latter scenario, major widespread floods in Great Britain (GB), observed during December 1979, October 2000, December 2002–January 2003, November-December 1992 and January–February 1995, were mostly driven by cyclonic and westerly LWTs [25]. Others have used LWTs to reconstruct the synoptic drivers of fluvial floods in GB since the 1870s [49]. Furthermore, some work uses LWTs to quantify changes in the strength of the *nocturnal Urban Heat Island (UHI)*—a phenomenon that may be associated with combined heatwave and air pollution events within cities [38,50], and is mainly driven by anticyclonic weather patterns. The LWTs classification scheme, although initially developed for the UK [25,36,45,51–63], was also recently applied in other mid-latitude regions, for example Sweden [64,65], the Iberian Peninsula [66,67] and Spain [68,69]. As far the authors are aware, no study has yet investigated links between LWTs and multi-hazards in AOGCMs projections up to 2100. Such an assessment could raise awareness of risks thereby informing resilience and disaster risk reduction measures, from local to regional scales.

Previous evaluations for Europe and the BI show that Coupled Model Intercomparison Project Phase 5 (CMIP5) AOGCMs generally reproduce LWTs, calculated using daily sea-level pressure (SLP) fields, but there are recognized biases [53,54]. For example, CMIP5 AOGCMs are not yet able to simulate correctly the number of anticyclonic (A-type) patterns and hence blocking episodes, with the former being underestimated in northern Europe and the BI, but overestimated in southern Europe [53,54,70]. Other biases are found for cyclonic (C-type) and westerly (W-type) occurrences, with both being overestimated across Europe [54]. These studies also examined future changes in frequency of LWTs and blocking episodes by comparing historical conditions with Representative Concentration Pathway (RCP) 8.5, to determine how such changes might affect European temperatures. The A-type is projected to increase significantly over the BI during all seasons except for winter December-January-February (DJF), the C-type to decrease in all seasons, and the W-type to increase except in summer June-July-August (JJA) by the end of the century [54]. Overall, blocking episodes are projected to decrease for the BI in DJF and JJA by 2061–2090 (RCP8.5) [70].

We extend these analyses by assessing the ability of a CMIP5 [71] multi-model sub-ensemble (MME) of 10 AOGCMs at reproducing historical seasonal persistence of daily LWTs over the BI [33–36]. We define 2-day persistence as the probability that a given LWT will occur on any two successive days. Climate model simulations of historic LWTs are compared with those derived from the 20th Century (20CR) [72], National Center for Environmental Prediction (NCEP) [73] reanalyses, and Lamb's catalogue of subjectively defined weather types [33,74]. We investigate how persistence and seasonal frequencies are projected to change within the full 21st century under RCP8.5 and RCP4.5, with persistence assessed for both the MME mean (MMEM) and individual AOGCMs. We also quantify and discuss the implications of future multi-hazards, here identified as nearly concurrent multi-basin fluvial flooding and ETCs impacting GB in winter [25] or combined summer heatwave and poor air quality events over London [38]. Thus, two multi-hazard metrics are applied, along with their evaluation under RCP8.5 and RCP4.5 projections up to 2100. These are the likelihood of (1) multi-basin fluvial flooding linked with ETCs (*F-Score*) and (2) changing intensity of the nocturnal UHI.

## 2. Methods and Data

### 2.1. Lamb Weather Types (LWTs)

Daily SLP patterns are categorized using the system of LWTs [33] via an objective classification scheme centred over the BI (Figure 1) [34,35]. Choice of the LWTs objective scheme is justified by the fact that this methodology and weather typing classification was originally developed for the BI. LWTs of similar airflow properties are derived from a 5° by 10° latitude-longitude grid array (Figure 1) and computed from daily (12 UTC) SLP values at each grid point. The airflow characteristics are expressed by the following set of equations, where the SLP integers' subscripts correspond to the grid point reference numbers in Figure 1:

$$W = \frac{1}{2}(SLP_{12} + SLP_{13}) - \frac{1}{2}(SLP_4 + SLP_5) \text{ (westerly flow)} \tag{1}$$

$$S = 1.74\left[\frac{1}{4}(SLP_5 + 2.0 \times SLP_9 + SLP_{13}) - \frac{1}{4}(SLP_4 + 2.0 \times SLP_8 + SLP_{12})\right] \text{ (southerly flow)} \tag{2}$$

$$F = \left(S^2 + W^2\right)^{1/2} \text{ (resultant flow)} \tag{3}$$

$$ZW = 1.07\left[\frac{1}{2}(SLP_{15} + SLP_{16}) - \frac{1}{2}(SLP_8 + SLP_9)\right] - 0.95\left[\frac{1}{2}(SLP_8 + SLP_9) - \frac{1}{2}(SLP_1 + SLP_2)\right] \text{ (westerly shear vorticity)} \tag{4}$$

$$ZS = 1.52\left[\begin{array}{c}\frac{1}{4}(SLP_6 + 2.0 \times SLP_{10} + SLP_{14}) - \frac{1}{4}(SLP_5 + 2.0 \times SLP_9 + SLP_{13}) - \frac{1}{4}(SLP_4 + 2.0 \times SLP_8 + SLP_{12}) \\ + \frac{1}{4}(SLP_3 + 2.0 \times SLP_7 + SLP_{11})\end{array}\right] \text{ (southerly shear vorticity)} \tag{5}$$

$$Z = ZW + ZS \text{ (total shear vorticity)} \tag{6}$$

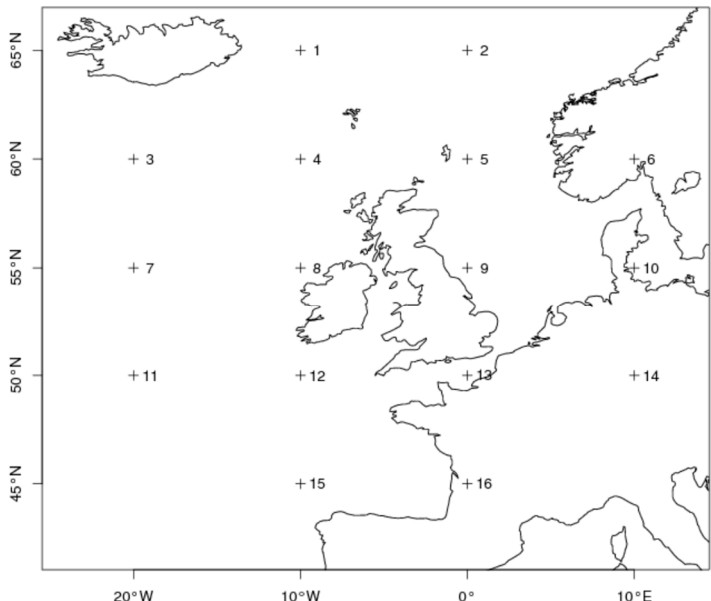

**Figure 1.** Grid points used to calculate Jenkinson flow and vorticity terms for the British Isles (BI). Numbers refer to those points used in Equations (1)–(5).

Flow units are derived from the geostrophic approximation (each equivalent to 1.2 knots) and they are, along with the geostrophic vorticity units, expressed as hPa per 10° latitude at 55° N (100 units are equivalent to $0.55 \times 10^{-4} = 0.46$ times the Coriolis parameter at 55° N). Three coefficients are used within Equations 2, 4 and 5 to account for variations in relative grid spacing at different latitudes with latitude ($\psi$) here set as 55° [34]: $S$ is multiplied by 1.74, derived from $1/\cos(\psi)$; $ZW$, 1.07 and 0.95 from $\sin(\psi)/\sin(\psi - 5°)$ and $\sin(\psi)/\sin(\psi + 5°)$; $ZS$, 1.52 from $1/2(\cos(\psi))^2$.

The last step for defining LWTs is to apply five rules [33–35]:

1. Flow direction is given by $\tan^{-1}(W/S)$ and is calculated on an eight-point compass with 45° per sector. If W is positive, add 180°. Thus, the W-type occurs between 247.5° and 292.5° (Equations (1) and (2));

2. Lamb pure directional weather types (e.g., N, S, or E-types) correspond to an essentially straight flow, when $|Z|$ is less than $F$ (Equation (6));

3. Lamb's pure cyclonic (C) and anticyclonic (A) types are identified when $|Z|$ is greater than $2F$, respectively with $Z > 0$ and $Z < 0$ (Equations (3) and (6));

4. Lamb's hybrid types (e.g., AE and CSW) are characterised by a flow partially anticyclonic/cyclonic, with $|Z|$ lying between $F$ and $2F$ (Equations (3) and (6));

5. An unclassified (U) type is obtained when $F$ and $|Z|$ are less than 6, with the choice of 6 depending on grid spacing, meaning that if using a grid resolution finer than 5° by 10° latitude-longitude it needs to be tuned (Equations (3) and (6)).

The objective classification scheme yields 27 LWTs comprised of two synoptic types (anticyclonic A and cyclonic C), five purely directional types (westerly W, north-westerly NW, easterly E, northerly N, and southerly S), 19 hybrid combinations of synoptic and directional types (e.g., CNW, CSE and AE), and 1 unclassified (U) type (Table 1) [33–35,75]. For persistence and frequency analyses, we focus on the seven synoptic and directional LWTs plus the U-type; counts of hybrid types were spread across the main types as per Lamb's original definition [33,76] and common practice within earlier studies [35–37,77]. We assess LWT persistence and frequency for summer (JJA), autumn (September–October–November, SON), winter (DJF) and spring (March–April–May, MAM). When

calculating indices of future multi-hazards, hybrid LWTs were not incorporated into the seven main types as the F-Score and nocturnal UHI indices require these individual weather patterns to be considered independently. For a more detailed description with maps showing the pressure patterns associated with the main LWTs we refer the reader to [33,34].

**Table 1.** Description of the seven main Lamb Weather Types (LWTs) and unclassified (U) type [33,75].

| LWT | Description |
|---|---|
| Anticyclonic (A) | Anticyclones centred over, near, or extending over the British Isles. |
| Cyclonic (C) | Depressions passing frequently or stagnating over the British Isles. The central isobar of the depression should extend over the mainland of Britain or Ireland. |
| Westerly (W) | High pressure to the south and low pressure to the north, giving a sequence of depressions travelling eastward across the Atlantic. This is the main, progressive zonal type. |
| North-westerly (NW) | Azores anticyclone displaced northeast or north towards the British Isles. Depressions forming near Iceland and travelling south-east into the North Sea. |
| Easterly (E) | Anticyclones over Scandinavia extending towards Iceland across the Norwegian Sea. Depressions generally to the south of the region over south-west Europe and the western Atlantic. |
| Northerly (N) | High pressure to the west or northwest of Britain extending from Greenland southwards, possibly as far as the Azores. Depressions travel southward from the Norwegian Sea. |
| Southerly (S) | High pressure over central and northern Europe. Depressions blocked to the west or travelling north or north-eastwards off western coasts. |
| Unclassified (U) | Weather pattern weak or chaotic. |

*2.2. Data*

Weather patterns were derived from the SLP produced by each AOGCM in our CMIP5 MME listed in Table 2 [71]. CMIP5 data were obtained from the World Climate Research Programme (WCRP, https://esgf-node.llnl.gov/projects/cmip5/). We defined the historical period as the 1980s (1971–2000) whereas the future was divided into three 30-year periods: the 2020s (2011–2040), 2050s (2041–2070) and 2080s (2071–2100). Such subdivision of time-periods is common practice within the climate modelling community, e.g., [20,78,79], as it allows us to evaluate information belonging to four 30-year periods up to 2100. We note that CMIP5 observational runs are available from 1950–2005 and future RCP runs cover the period 2006–2100. The CMIP5 AOGCMs and MMEM outputs for the historical period were compared with LWTs derived from 20CR [72], NCEP [73] reanalyses and Lamb's subjective catalogue, which ends in 1997 and was based on observed daily surface and mid-troposphere (500 mb) pressure charts at noon [33,74]. The 20CR reanalysis product is derived by making use of synoptic surface pressure observations. This has a spatial resolution of $2° \times 2°$ (latitude $\times$ longitude) and covers the 1871-present period with 6 h time steps and 28 pressure levels [72]. On the other hand, NCEP reanalysis is computed from a different set of observations (e.g., land surface, ship, aircraft and satellite), and covers the period 1948-present with $2.5° \times 2.5°$ (latitude $\times$ longitude) spatial resolution, 6h time steps and 17 pressure levels [73]. Both 20CR and NCEP datasets are largely used for climate model evaluations and their biases can be summarised as follows: (i) 20CR overestimates cloud fraction and precipitation [80]; and (ii) NCEP underestimates the temperature, overestimates the wind-speed and monthly precipitation variability [81]. The MMEM was built by first deriving the LWTs and their seasonal persistence and frequencies in each AOGCM, then averaging these metrics within each

time-period. The choice of the models included in our MME (Table 2) reflects a range of research institutes running similar boundary forcing experiments.

**Table 2.** Atmosphere-Ocean General Circulation Models (AOGCMs) multi-model sub-ensemble (MME) used in the analyses.

| Model Name | Research Institute | Lat-Lon Resolution | Ensemble Member |
|---|---|---|---|
| HadGEM2-ES | Met Office, United Kingdom | $1.25° \times 1.875°$ | r1i1p1 |
| MPI-ESM-LR | Max Planck Institute for Meteorology, Germany | $1.9° \times 1.9°$ | r1i1p1 |
| MRI-CGCM3 | Meteorological Research Institute, Japan | $1.1° \times 1.1°$ | r1i1p1 |
| CNRM-CM5 | National Centre for Meteorological Research, France | $1.4° \times 1.4°$ | r1i1p1 |
| CanESM2 | Canadian Center for Climate Modeling and Analysis, Canada | $2.8° \times 2.8°$ | r1i1p1 |
| MIROC5 | Model for Interdisciplinary Research on Climate, Japan | $1.4° \times 1.4°$ | r1i1p1 |
| CSIRO-Mk3.6.0 | Commonwealth Scientific and Industrial Research Organisation, Australia | $1.9° \times 1.9°$ | r10i1p1 |
| IPSL-CM5A-LR | Institute Pierre-Simon Laplace, France | $1.9° \times 3.75°$ | r1i1p1 |
| CCSM4 | National Center for Atmospheric Research, USA | $0.94° \times 1.25°$ | r6i1p1 |
| GFDL-CM3 | Geophysical Fluid Dynamics Laboratory, USA | $2° \times 2.5°$ | r1i1p1 |

The columns in Table 2 show the: (1) AOGCM name; (2) research institute where the model was developed; (3) resolution for latitude by longitude in degrees; and (4) ensemble member analysed. For all models the historical and Representative Concentration Pathway (RCP) 8.5 (and RCP4.5) sea-level pressure (SLP) outputs are used to calculate daily LWTs for the British Isles (BI).

### 2.3. Persistence and Trend Analyses

Weather pattern persistence is defined here as the conditional probability ($p_{jj}$) that a given LWT$_j$ on day($t$) is followed by the same LWT$_j$ on day($t$ + 1) [82,83]. This diagnostic was extracted for the 7 main LWTs and the U-type using the diagonal cells of Markov-chain transition matrices. This enabled estimation of historical (1980s) and future (2020s, 2050s, and 2080s) seasonal persistence for the MMEM as well as for individual AOGCMs for impactful weather types and seasons, the 20CR, NCEP reanalyses and Lamb's subjective catalogue.

Uncertainty in $p_{jj}$ for the 1980s was calculated by boot-strapping (n = 1000) 30-year seasonal simulations using the *markovchain* package within the R framework [84]. This algorithm stochastically generates *n* series of daily LWTs from the original conditional distributions of the weather patterns in each AOGCM, then recomputes $p_{jj}$ from each series. The resulting $p^{BOOTSTRAP}_{jj}$ is the mean of all $p_{jj}$ across the 1000 series, for each AOGCM. The 95% confidence intervals of $p^{BOOTSTRAP}_{jj}$ are obtained from the cumulative distribution of the 1000 values of $p_{jj}$ for each AOGCM.

Statistical significance of changes in persistence for the AOGCM sub-ensemble between the 1980s and future periods (Tables S1 and S2) was assessed using a Mann-Whitney-Wilcoxon two-tailed test [85] applied to the 10 estimates of $p^{BOOTSTRAP}_{jj}$ for each time period. Changes in $p_{jj}$ between the 1980s and future periods for *individual* AOGCMs were regarded as statistically significant if future persistence of a given LWT and AOGCM fell outside the 95% confidence intervals of the $p^{BOOTSTRAP}_{jj}$ range of that AOGCM for the 1980s.

To detect both linear and non-linear annual changes in the total seasonal counts of LWTs MMEM frequencies under RCP8.5 and RCP4.5 scenarios, a trend analysis was performed for the entire 2006-2100

time-period. For illustrative purposes, we only show trends for anticyclonic (A, summer JJA), cyclonic (C, autumn SON) and westerly (W, winter DJF) types as indicators of impactful weather across the BI. Results are also presented for the southerly (S, spring MAM) types as this LWT shows most significant changes in seasonal persistence according to the non-parametric Mann-Whitney-Wilcoxon two-tailed test between the 1980s and each of the three future periods (i.e., 2020s, 2050s and 2080s). A modified Mann-Kendall test, which takes into account possible autocorrelation within the time series, was applied to both RCP8.5 and RCP4.5 seasonal MMEM LWTs frequencies [86]. The significance of trends, along with their relative Sen's slopes, are shown in Table S3 [87].

### 2.4. Indices of Winter Fluvial Flooding-Wind Hazards and Summer UHI Intensity

As a measure of concurrent fluvial flooding-wind hazards we calculated an extended version of the F-Index [25,49], here defined as the F-Score, for each AOGCM, MMEM, 20CR, NCEP and Lamb's subjective catalogue, covering the 1980s, 2020s, 2050s, and 2080s, for selected LWTs known to drive these multi-hazard events [25] during winter under both RCP8.5 and RCP4.5. The F-Index is the ratio of observed to expected frequency of fluvial floods for a given LWT, where values greater than 1 show higher than expected likelihood. Ten LWTs are known to be associated with historic, multi-basin fluvial floods [25], of which eight (C, CS, CSW, CNW, S, SW, W, and NW-types) increase their likelihood and two (N and A-types) reduce likelihood. All other LWTs are weighted zero. The F-Score, for each AOGCM, is then calculated by multiplying the winter DJF frequencies ($freq\_djf_j$) of these LWTs by their $F\_Index_j$ (as per Event Set E in [25]) and by summing these values:

$$F\_Score_i = \sum_{j=1}^{10} freq\_djf_{i,j} \times F\_Index_{i,j} \qquad (7)$$

where *i* represents the single AOGCM, 20CR, NCEP and Lamb's subjective datasets within the relative time periods of 1980s, 2020s, 2050s, 2080s and *j* is the given LWT considered from the 10 types mentioned above. The higher the F-Score, the greater the likelihood of concurrent multi-basin fluvial flooding and wind hazards within winter, over the specified time horizon and RCP scenario.

As a proxy for combined heatwave and poor air quality hazards occurring during summer, we use observed, simulated and projected nocturnal UHI temperatures in tenths of degree Celsius for London (UK) [38], using the same datasets, time periods and RCPs as per the F-Score. The UHI phenomenon is caused by absorption and trapping of heat as well as by changed airflows and sensible heat fluxes within the built environment. The simplest form of UHI metric (used by [38]) is based on the daily temperature difference between an urban and rural reference site (during daylight or night hours). These values may then be stratified by LWT to show the extent to which some weather patterns favour extreme UHI episodes. Previous studies show that the anticyclonic (A) weather types are associated with extreme UHI events [38,50]. The UHI metric, for each AOGCM, was derived as follows by: (i) multiplying LWT summer JJA frequencies ($freq\_jja_h$) by their respective average UHI intensities taken from [38] ($UHI\_w_h$); (ii) summing these values; and (iii) dividing the total from step (ii) by the number of days in the period analysed ($days_h$) to give the mean daily UHI intensity:

$$UHI_i = \sum_{h=1}^{27} \frac{freq\_jja_{i,h} \times UHI\_w_{i,h}}{days_{i,h}} \qquad (8)$$

where *i* is the same notation as per the F-Score and *h* refers to the 27 LWTs.

To assess the statistical significance of changes in the AOGCM output for the 1980s and future 2020s, 2050s and 2080s periods (for both the F-Score and nocturnal UHI temperatures) we applied a similar approach as per persistence. Here, n = 1000 boot-strapped samples of daily LWT series (based on conditional distributions for all seasons combined) were generated for each AOGCM run in the 1980s. Next, the F-Score or UHI were calculated for every series and AOGCM, then averaged and

confidence limits established as before. This procedure shows the extent to which estimates for the future indices fall within the 95% confidence range of the boot-strapped estimate for each AOGCM in the 1980s.

Sample sizes varied depending on the index and AOGCM. For the F-Score, we considered the period 1971–2001 to capture January and February of winter 2000/01. Here, models with leap years have a total of 11,323 days, models without leap years 11,315 days and the HadGEM2-ES model (with 360 days per year) has 11,160 days. For the UHI, the calendar years 1971–2000 were used as we are interested in summer temperatures, with leap year AOGCMs having 10,958 days, non-leap years models 10,950 days and the HadGEM2-ES 10,800 days.

Figure 2 provides a synthesis of the data and methodological framework.

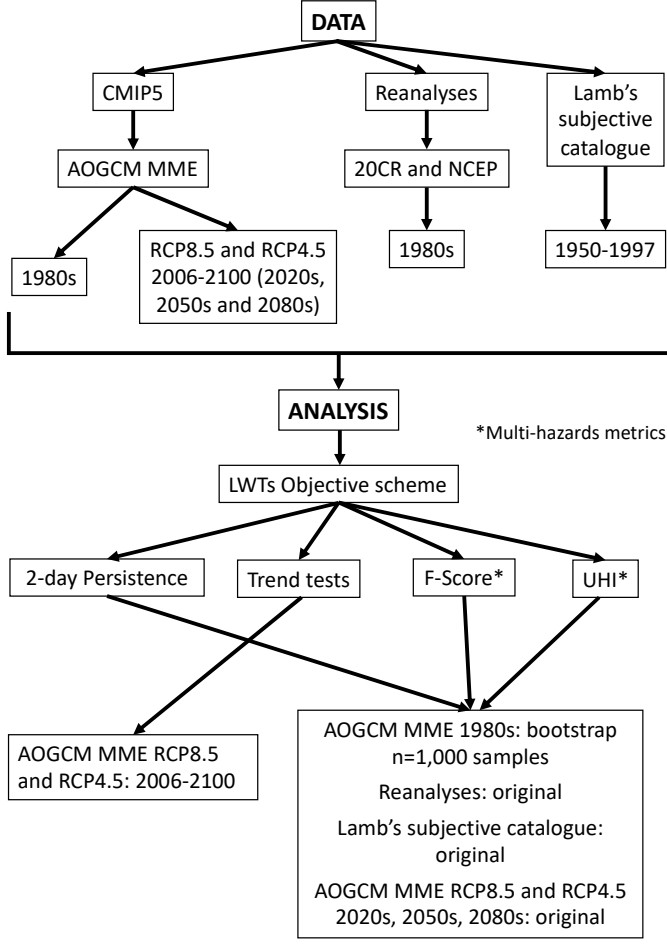

**Figure 2.** Main data and methodology steps. The figure synthesise the procedures described in Section 2. Methods and Data.

## 3. Results

### 3.1. Persistence of Weather Patterns (MME)

The A, C and W patterns are the most frequent weather types affecting the BI. Overall, the MME replicates weather type persistence during the four climatological seasons, when compared with 20CR [72] and NCEP [73] reanalyses for the historical period (1980s) (Figure 3). There is less agreement between Lamb's subjectively classified daily weather catalogue and both the MME and reanalyses. A-type persistence is more variable within the MME and on average underestimated in winter, consistent with previous studies [53,54]. There is closer agreement for the A-type in other seasons.

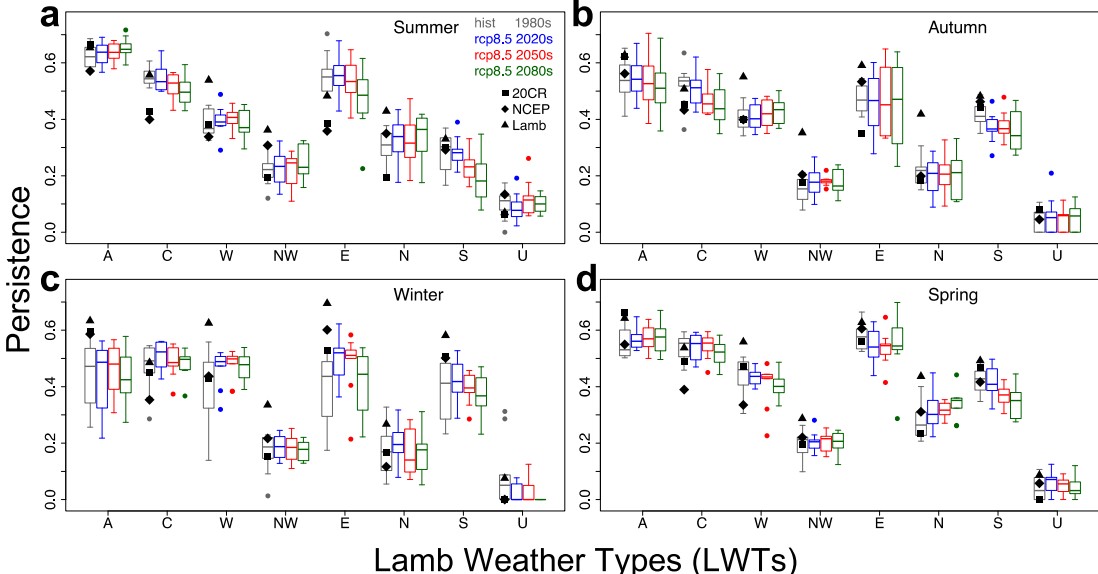

**Figure 3.** Persistence of the seven main Lamb Weather Types (LWTs) plus unclassified (U) type under Representative Concentration Pathway (RCP) 8.5. Persistence is calculated for (**a**) summer June-July-August (JJA), (**b**) autumn September–October–November (SON), (**c**) winter December–January–February (DJF) and (**d**) spring March–April–May (MAM), for the historical 1980s (1971–2000) and under RCP8.5 by the 2020s (2011–2040), 2050s (2041–2070) and 2080s (2071–2100). Boxplots show distributions of persistence in each LWT, for the 10-member Atmosphere-Ocean General Circulation Models (AOGCM) ensemble, compared with the 20th Century (20CR), National Center for Environmental Prediction (NCEP) reanalyses and the Lamb's subjective catalogue. Segments show the minimum, 1st quartile, median, 3rd quartile and maximum. Outliers are shown by dots.

W-type persistence agrees with the reanalyses but is always less than in Lamb's catalogue. C-type persistence is overestimated by the MME in all seasons when compared to reanalyses as reported before [54] for Europe more generally. Such biases in the C-type could be interpreted as exaggerating the likelihood of flooding in the MME compared with reanalyses [49].

Figure 3 shows that the distributions of persistence are asymmetrical (or skewed) around the MME means for many of the weather types and time periods. This characteristic suggests potentially large biases in the estimation of extreme events, if relying on a single AOGCM. Changes in weather type persistence between the ensembles of historical and future periods within RCP8.5 (Figure 3) are weakly significant (*p*-value < 0.1, Mann-Whitney-Wilcoxon two-tailed test) for the C-type in summer and autumn by 2080s; W-type in winter by 2050s; E-type in summer by 2080s and winter for the 2020s and 2050s; N-type in spring by 2050s and 2080s; and S-type in summer by 2080s, autumn in all periods and spring by 2050s and 2080s (Table S1).

Results for RCP4.5 show similar changes in persistence compared to RCP8.5, although they are smaller (Figure S1). In particular, the C-type is found to change significantly (*p* < 0.1) only in summer by the 2080s; the E-type in winter by the 2080s; the N-type only in spring by the 2080s; and the S-type in summer by the 2050s and spring also by the 2020s (Table S2).

### 3.2. Persistence of Weather Patterns (AOGCMs)

Figure 4 shows persistence for the same future periods but for each AOGCM in the MME compared with the reanalyses and Lamb's catalogue, for impactful weather types and seasons. Significance of changes was assessed against the boot-strapped confidence limits for the 1980s. Most model projections under RCP8.5 fall outside the 95% confidence intervals of historical persistence. A-type MMEM persistence increases during summer (Figures 3a and 4a); C-type persistence decreases in all seasons,

most markedly in summer and autumn (Figures 3 and 4b); W-type persistence does not change in winter but increases in autumn and decreases in spring (Figures 3b–d and 4c).

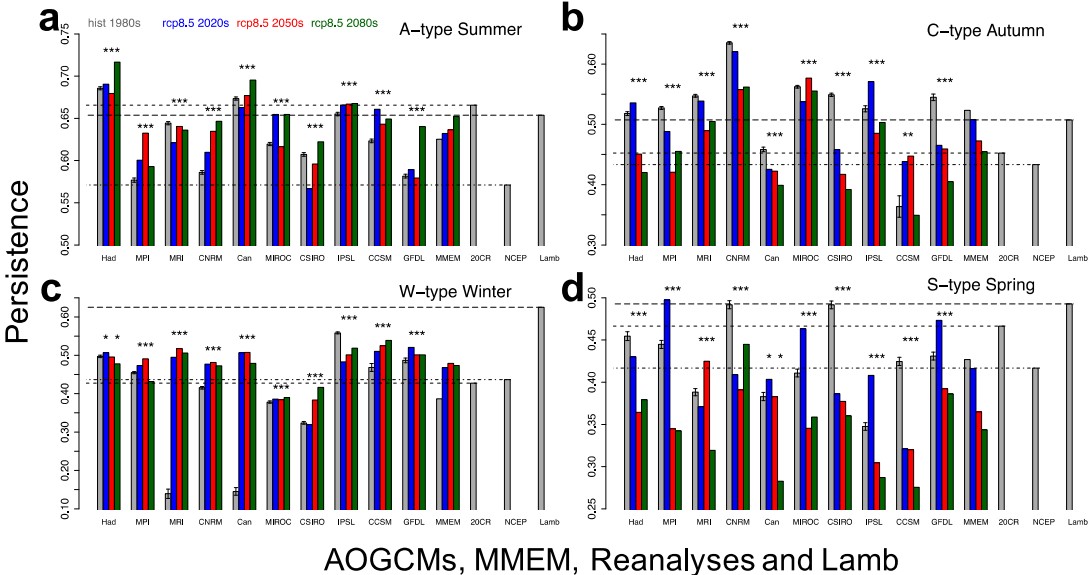

**Figure 4.** Persistence of selected LWTs and seasons for individual AOGCMs under RCP8.5. (**a**) Anticyclonic (A) type (summer JJA), (**b**) cyclonic (C) type (autumn SON), (**c**) westerly (W) type (winter DJF) and (**d**) southerly (S) type (spring MAM) in the 1980s compared with the 2020s, 2050s and 2080s under RCP8.5. Persistence is shown for individual AOGCMs alongside the multi-model ensemble mean (MMEM), 20CR, NCEP and Lamb's subjective catalogue. Asterisks (*) show model runs with persistence outside the 95% confidence intervals of the boot-strapped (n = 1000) estimates for the 1980s, shown here as black T-bars. Dashed lines represent the reanalyses and Lamb's catalogue values.

Amongst the other weather types, we note only a decrease in C- and E-types during summer, an increase in N-type in spring, and S-type persistence decreases in all seasons (Figures 3 and 4d). The AOGCMs showing the largest change in A-type persistence during summer are CNRM-CM5, GFDL-CM3 and MIROC5, with a significant increase of 0.06, 0.06 and 0.04 respectively between 1980s and 2080s. For the C-type in autumn, CSIRO-Mk3.6.0, GFDL-CM3 and HadGEM2-ES show a significant decrease in persistence, between 1980s and 2080s, of 0.16, 0.14 and 0.10 respectively. During winter, for the W-type, the AOGCMs showing the largest change, between the same 1980s and 2080s periods, are MRI-CGCM3, CanESM2 and CSIRO-Mk3.6.0 with a significant increase in persistence of 0.37, 0.33 and 0.09 respectively.

Analysis of RCP4.5 output shows similar, though less marked, results when compared to RCP8.5 (Figure S2). Under the lower emission scenario, we find that most AOGCMs project persistence that falls outside the 95% confidence intervals of the 1980s. A-type MMEM persistence in summer could increase slightly, in particular during the 2080s Figures S1a and S2a), C-type in autumn may decrease (Figures S1b and S2b), W-type during winter is projected to remain stable across the three future periods (Figures S1c and S2c) and S-type persistence in spring decreases by 2100 (Figures S1d and S2d). The C-type in summer and A-type in autumn exhibit decreased persistence, whereas the E-type shows a marked increase in persistence during winter; findings that differ from RCP8.5 (Figure S1).

### 3.3. Frequency of Weather Patterns (MMEM)

Projected frequency trends for selected weather types and seasons under RCP8.5 (2006–2100) are shown in Figure 5. Summer A- and winter W-type frequencies could rise significantly ($p < 0.01$, Table S3) by 0.8 and 0.2 days per decade respectively over the period 2006–2100. Conversely, C- and S-type frequencies decrease significantly ($p < 0.01$, Table S3) in autumn and spring respectively. No

significant trends are found for C-type frequency during winter. Sen's slopes for the MMEM with their statistical significance are given in Table S3 for each weather type, season and RCP. We also computed the Sen's slopes for A-type in each AOGCM during summer (RCP8.5, not shown here) to check whether the increase in A-type was solely due to a few models showing a large increase in this weather type. We found that all models within the MME show a positive increase in A-type frequency, with 7 out of 10 AOGCMs showing significance at the 90% level, with no outliers skewing the MMEM. Among other seasons (not shown), a significant decrease in annual frequencies is observed for the C-type during summer ($p < 0.01$) and spring ($p < 0.05$), along with a significant ($p < 0.01$) increase in A-type during spring, which all reflect the changes in persistence (Figure 3a,d).

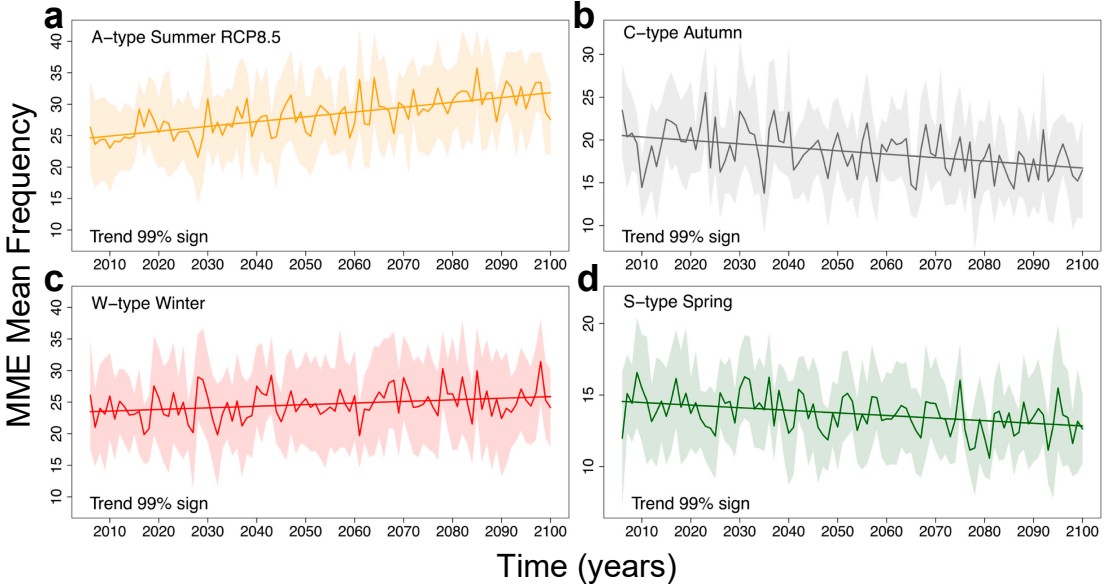

**Figure 5.** Projected annual frequencies for selected LWTs and seasons under RCP8.5. Frequencies are shown as MMEM for (**a**) A-type (summer JJA), (**b**) C-type (autumn SON), (**c**) W-type (winter DJF) and (**d**) S-type (spring MAM) LWTs under RCP8.5 (2006–2100). Trends are statistically significant at the 1% level (*p*-value < 0.01, modified Mann-Kendall test). Shaded areas represent the 95% confidence intervals of the MMEM. The trend lines refer to the Sen's slopes calculated with the modified Mann-Kendall test.

Projections of MMEM frequencies for the same LWTs and seasons but under RCP4.5 are shown in Figure S3 and Table S3. Results for RCP4.5 reflect the scenarios of RCP8.5 although the Sen's slopes are less extreme and statistically significant. The A-type frequency is projected to increase significantly ($p < 0.01$, Figure S3a and Table S3) during summer, C-type in autumn is set to decrease ($p < 0.05$, Figure S3b), W-type frequency in winter shows no significant trend (Figure S3c), and the S-type during spring decreases significantly ($p < 0.05$, Figure S3d). As per RCP8.5, we also observe (not shown) a significant decrease in C-type frequencies during summer ($p < 0.01$) and spring ($p < 0.05$) and an increase in the A-type during spring ($p < 0.05$), matching the relative changes in persistence (Figure S1a,d).

### 3.4. Application to Future Multi-Hazards

In Figure 6, we extend an earlier analysis [25] based on impactful LWTs found to generate concurrent fluvial flooding-wind hazards in GB (see Section 2.4). Thus, the F-Score for each single AOGCM, MMEM, 20CR, NCEP and Lamb's subjective datasets and 1980s, 2020s, 2050s and 2080s time periods are shown for winter DJF weather patterns under RCP8.5. The F-Score is a measure of the severity of future concurrent fluvial flooding-wind hazards, such that higher values represent more severe impacts compared to lower ones. Here, we show that the baseline risk from multiple flood-wind hazards is overestimated by all but two of the AOGCMs (HadGEM2-ES and MIROC5)

when compared to NCEP, 20CR reanalyses and Lamb's subjective catalogue for the 1980s. Assuming the same bias holds in the future, the AOGCMs evaluated here likely overestimate *absolute* future risk from concurrent flood-wind hazards by 2100. Moreover, in a similar way as per Figure 4, there exists a large variability between the AOGCMs, so F-Score results are mixed with some AOGCMs suggesting increased/decreased risk of flood-wind hazards by the end of the 21st century. Lastly, by looking at the MMEM we conclude that, although overestimated by AOGCMs, future risk from concurrent flood-wind hazards could increase by 2100 compared with the 1980s. Among the AOGCMs, those showing the largest F-Score increase between the 1980s and 2080s are CanESM2, CCSM4 and IPSL-CM5A-LR. Results for RCP4.5 are shown in Figure S4 and they agree with what was found for RCP8.5, with large variability amongst AOGCMs and MMEM F-Score even slightly higher than RCP8.5.

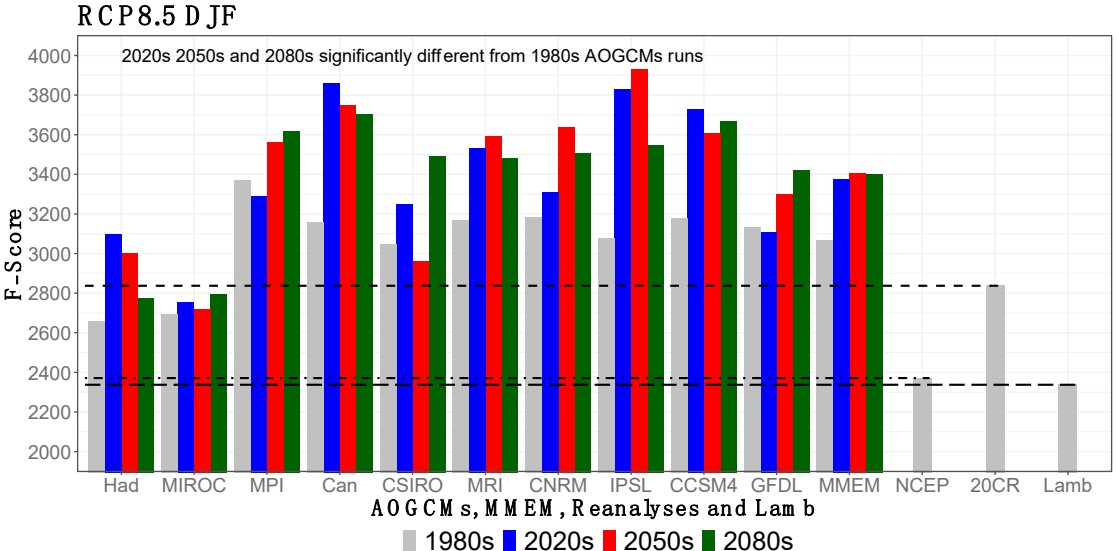

**Figure 6.** F-Score for LWTs associated with concurrent fluvial flooding-wind hazards during winter (DJF). The F-Score is shown for each AOGCM, MMEM, NCEP, 20CR and Lamb's subjective catalogue for the 1980s, 2020s, 2050s and 2080s periods. The LWTs used for calculating the F-Score are associated with concurrent multi-basin fluvial flooding and wind hazards within Great Britain (GB) [25]. The 1980s MME F-Score were estimated from the mean of n = 1000 boot-strapped samples and all the future 2020s, 2050s and 2080s periods are significantly different from these, as the F-Score of the latter fall outside the 95% confidence intervals of the 1980s means. The AOGCMs 1980s confidence intervals bars are not shown for simplicity because they are vanishingly narrow. Dashed lines represent the reanalyses and Lamb's catalogue values.

Summer nocturnal UHI temperatures in tenths of °C for London (UK), were estimated for RCP8.5, by using UHI values obtained in a previous study [38] (Figure 7 and Section 2.4). Our results show that AOGCMs replicate nocturnal UHI temperatures, although there is a tendency for underestimation by the majority of AOGCMs except HadGEM2-ES and MIROC5 which show good agreement when compared to 20CR, NCEP and Lamb's subjective catalogue as per the F-Score (Figure 6). We also note that there is less variability within the MME than displayed in Figures 4 and 6. Lastly, almost all the AOGCMs and MMEM show a statistically significant increase in UHI by the end of 2100, that could translate into an increased multi-hazard risk from heatwave and poor air quality events associated with persistent A weather types [38,55,88,89]. The projected increase in the MMEM UHI between the 1980s and 2080s is 0.15 °C under RCP8.5. The AOGCMs that show the largest increase in nocturnal UHI temperatures between 1980s and 2080s are CanESM2, HadGEM2-ES and CCSM4 with respectively 0.23, 0.22 and 0.22 °C. Results for RCP4.5 agree with the RCP8.5 projections although the changes are less marked (Figure S5). Implied increases in the risk of urban air pollution hazards are potentially conservative given policies to phase out conventional cars in the UK by 2050.

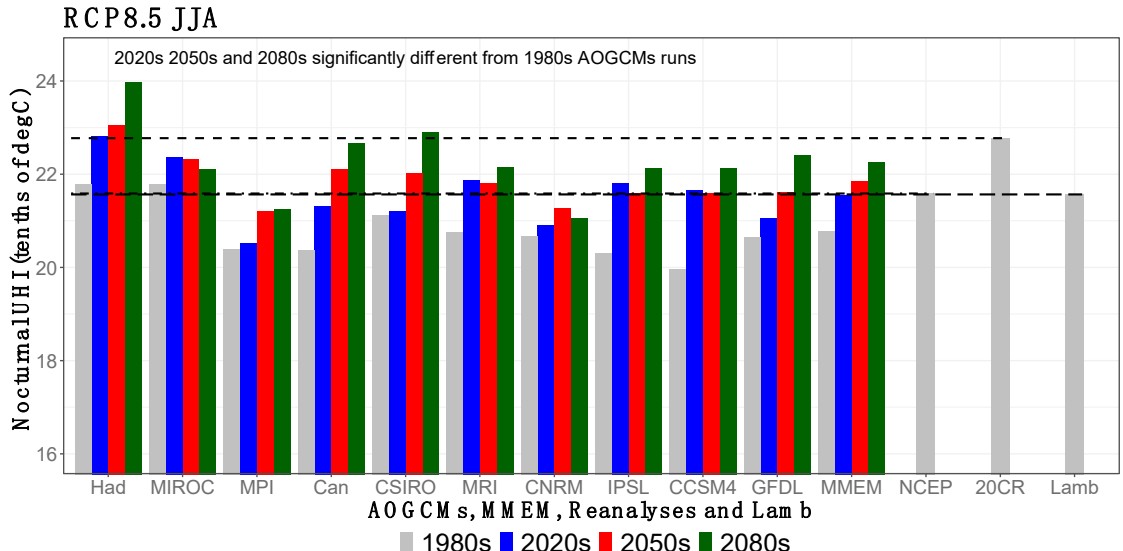

**Figure 7.** As per Figure 6 but for London's nocturnal Urban Heat Island (UHI) in tenths of degree Celsius (°C) during summer (JJA).

## 4. Discussion and Conclusions

As found in our analysis, greater A-type persistence and frequency during summer likely implies more blocking episodes with increased risk of poor air quality, drought and heatwaves [1,5,90,91]. A growing number of studies propose physical mechanisms that link Arctic Amplification (AA) [92] to more persistent weather patterns, which in turn enhance the likelihood of extreme weather events in the northern hemisphere mid-latitudes. The AA may affect the polar jet stream by making Rossby waves more meridional (or wavier) and by weakening its flow. A wavier and weaker jet stream in summer favours more persistent extreme weather and it is also thought to extend ridges northward, enhancing such effects [1–3,90,91,93–95]. In contrast, another study suggests that increasing trends in meridional extent of the jet stream, along with blocking events, may be an artefact of the methodologies used [85].

Our results support earlier analysis [54], and are consistent with the proposed mechanisms linking *observed* AA with mid-latitude weather extremes. On the one hand, AA could have limited effect on simulated CMIP5 blocking over Eurasia under RCP8.5 in the second half of the 21st century [96]. Other work, that makes use of three different algorithms for computing blocking (i.e., anomaly, absolute and hybrid methods) also shows an overall decrease in CMIP5 blocking events over the BI in winter DJF and summer JJA, during 2061–2090 (RCP8.5) [70]. Our findings for anticyclonic weather appear to contradict this. Although A-type persistence and frequency are equivalent to blocking *per se*, we would expect the studies to agree as both mechanisms involve high pressure weather systems. A common denominator between our findings and studies of blocking [70,97] is the underestimation of A-type/blocking events by CMIP5 models. However, further research is needed to reconcile apparently contradictory findings. Possible explanations are that results depend on the exact spatial domain and/or suite of AOGCMs analysed in each MME, as well as on the methodology used to define A-type days and blocking events.

In our study, less persistent C-types in autumn suggests lower likelihood of heavy rainfall, with reduced recharge of soil moisture and aquifers at the start of the hydrological year, thereby favouring winter droughts. Fewer cyclonic days may also translate into less frequent severe gales and fluvial flooding episodes [49], as in GB extreme multi-basin fluvial flooding events are strongly associated with C-type weather over time windows from 1 to 19 days [25]. Conversely, more frequent zonal airflow (W-type) in winter may counteract some loss of precipitation from the C-type, especially across higher elevation regions of the north and west BI where there is strong orographic enhancement [98].

Such changes may also be attributed to AA, however, the physical mechanisms linking AA to changes in northern hemisphere mid-latitude circulation currently remains an open question.

From our analyses it is also possible to infer future changes with respect to multi-hazards [15,17], through the F-Score and nocturnal UHI temperatures. Recent analyses show that in GB nearly concurrent multi-basin fluvial flooding and extreme wind events are driven by selected LWTs mainly associated with C- and W-types [25]. These multi-hazard events can generate significant economic losses hence projections of such events may help in evaluating future risks and in improving resilience. We show that during winter DJF our ensemble of AOGCMs overestimate the F-Score when compared to 20CR, NCEP reanalyses and Lamb's subjective dataset. Even so, by the end of 2100 the MMEM shows a statistically significant increase in the F-Score compared with the 1980s within those same models, suggesting that the risk of concurrent fluvial flooding-wind impacts may become more severe in a warmer world. The two AOGCMs that show the closest agreement with the reanalyses are HadGEM2-ES and MIROC5.

Our results for nocturnal UHI temperatures in London modelled by AOGCMs agree with 20CR, NCEP and Lamb's subjective datasets, although they are slightly underestimated for the 1980s. As per the F-Score, HadGEM2-ES and MIROC5 are the AOGCMs that best represent the reanalyses and, therefore, they may be preferred when assessing these two multi-hazard scenarios. Nocturnal UHI severity could increase by 2100 under RCP8.5 (MMEM). Our results confirm an increasing trend of ~0.3 °C in nocturnal UHI in London found in an earlier study over the observational period 1950-2006 [38]. Our findings are also in line with the UK Climate Projections Science Report 2009 [99] which suggests that intense UHI events are highly correlated with A-type weather patterns, and that in London, intense UHI summer events could become more severe in the future [50]. However, further analysis of projections of UHI is needed with a larger AOGCM ensemble to better account for uncertainty. Our results for UHI also assume an unchanging urban landscape and pattern of artificial heat sources. Nevertheless, the present findings, when viewed as a significant increase in persistence and frequency of A-type weather pattern, suggest more favourable conditions for heatwaves and poor air quality events in London that could negatively impact human health [38,50,55,88,89].

Finally, we have illustrated how changes in the persistence and frequency of weather patterns are useful diagnostics of climate model realism and can translate into regional to local weather and climate risks scenarios, which could be helpful for developing narratives for decision-makers. However, caution needs to be taken when qualitatively converting synoptic weather pattern changes into local variability because AOGCM skill in reproducing climatic variables at local scales varies significantly and is not always consistent with observations. This is particularly true for precipitation where, for example, pressure fields alone are not able to provide reliable local projections [43]. In our work, the two reanalyses products and Lamb's subjective catalogue show different results. Thus, it is difficult at this stage to suggest a preferred observational dataset for AOGCM validation. However, the objective classifications have the advantage of consistency over the subjective Lamb's catalogue. Our suggestion, therefore, would be to use a large ensemble of open source reanalyses products, to better account for uncertainty coming from products with different characteristics.

With the UK Climate Projections 2018, now partly released and work underway for the third UK Climate Change Risk Assessment, weather pattern analysis could help to both evaluate the new projections and offer ways of explaining changes that are intelligible to a range of user communities. Similar links to persistence could be made in other regions with established weather pattern typologies, such as the *Grosswetterlagen* for Europe [100], hydrologically important weather types in the contiguous United States [101] and Spatial Synoptic Classification for North America [102].

**Supplementary Materials:** The following are available online at http://www.mdpi.com/2073-4433/10/10/577/s1, Figure S1: As per Figure 3 but for RCP4.5, Figure S2: As per Figure 4 but for RCP4.5, Figure S3: As per Figure 5 but for RCP4.5, Figure S4: As per Figure 6 but for RCP4.5, Figure S5: As per Figure 7 but for RCP4.5. Table S1: MME statistical significance of LWTs persistence for RCP8.5, Table S2: The same as Table S1 but for RCP4.5, Table S3: Sen's slopes of MMEM seasonal LWTs frequencies for RCP8.5 and RCP4.5.

**Author Contributions:** Conceptualization, P.D.L. and R.W.; methodology, P.D.L., R.W. and C.F.; software, C.H. and P.D.L.; formal analysis, P.D.L.; data curation, P.D.L. and C.H.; writing—original draft preparation, P.D.L.; writing—review and editing, P.D.L., R.W., J.H., C.F. and G.L.; supervision, R.W., J.H. and G.L.

**Funding:** P.D.L. was funded by a Natural Environment Research Council studentship awarded through the Central England NERC Training Alliance (CENTA http://www.centa.org.uk/; Grant No. NE/L002493/1) and by Loughborough University. CF was supported by the Collaborative Research Centre TRR 181 "Energy Transfer in Atmosphere and Ocean", funded by the Deutsche Forschungsgemeinschaft (DFG, German Research Foundation https://www.dfg.de/en/)—Projektnummer 274762653. The APC was funded by CENTA NERC.

**Conflicts of Interest:** The authors declare no conflict of interest. The funders had no role in the design of the study; in the collection, analyses, or interpretation of data; in the writing of the manuscript, or in the decision to publish the results.

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
