# Peer review of "Past and Projected Weather Pattern Persistence with Associated Multi-Hazards in the British Isles"

_atmosphere, doi:10.3390/atmos10100577_

Round 1

Reviewer 1 Report

Manuscript number: Atmosphere

Title: Past and projected weather pattern persistence with associated multi-hazards in the British Isles

General comments

I reviewed the article titled “ Past and projected weather pattern persistence with associated multi-hazards in the British Isles” and I observed that the manuscript investigates an important concern of multi-hazards in BI. If I understood correctly, the authors defined multi-hazards as a co-occurring (spatially or temporally) natural hazards. This study further highlights that their impacts could be different what a single natural hazard must be thought of and hence it is indeed necessary to make an effort in this direction. In my opinion, such a study is of great significance, given the growing threat of global warming, climate change and multi-hazards around the world. The paper entirely covered the scientific topics of Atmosphere journal. I am convinced with the results and conclusions however, state-of-the-art and methodology need improvement. Overall, the article in the present format has space to improve especially in making the methodology well-structured and clear. Furthermore, it would be great if the author(s) can answer a few of my concern raised below.

Specific comments

Scientific community around the world already addressing the concern of compound natural extreme events since decades. How about using the same terminology in this article rather multi-hazards? Isn’t multi-hazards might be misleading in the present scenarios? Are all the studies mentioned about multi-hazards (P2/L46-58) used the same definition of multi-hazards?

The authors give some vague/generic statements which need to be carefully rewritten/ omitted. For e.g. P2/L55-56, until recently, natural hazards….This is not true.

I somehow missed the motivation of the study? In the abstract authors mentioned that “we demonstrate that 2-day weather pattern persistence is a useful concept for both investigating climate risks from multi-hazard events as well as for assessing AOGCM realism (P1/L17-18) but the discussion on same/motivation is missing in the introduction. In addition, the claim 2-day patter persistence is a useful concept for both investigating climate risks from multi-hazards events as well as for accessing AOGCM realism” is globally true?

Also, I notice that in the abstract LWT is not there at all and it appears suddenly and the entire story revolves around it.

Data section has space for improvements. Kindly provide the source of the data and links to that. This indeed will be helpful to make the work reproducible. Further, “We defined the historical period as the 1980s (1971-2000) whereas the future was divided into three 30-year periods: the 2020s (2011-2040), 2050s (2041-2070) and 2080s (2071-2100).”Is it true that time period 2001-2018 is missing? If yes, why? Please clarify the section.

Further, P3/line 216-217, a trend analysis was performed for the 2006-2100 time period. I don’t understand this time period.

I strongly suggest inserting a methodology figure which describes the approach is steps.

Nevertheless, it was a great attempt and I motivate the team to work and explore more to fill the gaps. For example, what is missing here is the clear methodology throughout.

Reviewer 2 Report

The manuscript “Past and projected weather pattern persistence with associated multi-hazards in the British Isles” is assessing the performance of a selection of CMIP5 models to reproduce persistent weather patterns over Europe associated with severe weather. After validating the models using two reanalysis products and observations, the authors use the same CMIP5 models to predict future changes in these weather patterns including their persistence and future frequency of selected weather types. In a second step, the authors present changes in two specific indices that use major weather patterns as measures for changes in the occurrence of sea level pressure conditions that favour multi-hazard conditions under global climate change. The link to severe weather with potential socio-economic impacts makes the topic of this manuscript highly relevant and its link to multi-hazard events renders it timely. I recommend publication after minor revisions.

General comments:

Introduction/Methods

The authors aim to highlight the potential impact of the weather pattern they asses and their role in multi-hazard events. However, this link is not made very clear throughout the manuscript. I recommend including a paragraph in the introduction that highlights the link between the persistent weather patterns, or Lamb Weather Types (LWTs), to actual impacts from the historic period. Are there significant floods in the past that the authors could use as a standard example? Additionally, the authors could elaborate on which LWTs are the dominant contributors to the two indices that they are assessing.

Another concern is related to the authors use of the term flood-wind events as one example for a multivariate hazard. The use of flood in this context is very vague and could be interpreted differently depending on personal biases. Do the authors refer to pluvial, fluvial, or coastal flood hazard or all of them? The weather patterns for the different flood types are potentially quite different or, if the authors include all three types of flood hazards in this term, the flood part itself could already be a multivariate hazard in itself. This could be resolved by giving examples of events where the hazardous synoptic patterns caused a real-life event in the past and clearly stating which type of flood the authors refer to.

Further, the authors use Lamb Weather Types and present a very thorough analysis of the changes in the occurrence of these weather patterns.  However, they fail to properly introduce the concept of LWTs in the Introduction. A brief mentioning of the background of the concept of LWTs, their history, and the use of LWTs in current forecasting or risk assessments would be beneficial for readers that aren’t familiar with this concept. Figures of the different/main LWT conditions could be added as a supplementary Figure. Additionally, a better motivation as to why this metric was chosen over, for instance, Empirical Orthogonal Functions would improve the manuscript and help frame the choice of method.

Methods and Data

The introduction of the weather type and their calculations are somewhat confusing. The authors refer to different types, such as W-type, N, S, and E-type, without explaining what these types are. Adding a figure or table with a brief description of the main characteristics of the different weather types would help to clear this up for readers that are not familiar with Lamb Weather Types. The authors might consider changing the section introducing the five rules to define LWTs (lines 139 – 156) into a table which would help to introduce the main LWTs and their main characteristic airflow properties. If the authors decide against such a table, I suggest linking them to the equations a bit by referencing those equations involved in the different rules by number. For instance, rule 1 should include a reference to equations 1 and 2, etc. In rule 5, it is unclear how the choice of 6 is dependent on grid spacing. This needs to be elaborated in more detail for the reader to understand the concept.

To avoid confusion in the equations, instead of using bold integers to refer to the different grid points in Figure 1, I recommend using subscripts, e.g. SLP12 instead of 12 in equation 1.

When introducing the data sources, the authors fail to properly introduce the two reanalysis products. A brief description of the two and their differences, in particular caveats and shortcomings, should be included as they are used to validate the model results. Also, the authors should mention what measurements the observed Lamb Weather Type Catalogue are based on (satellite, in situ?).

Results

In the assessment of model performance, the author chose to focus on a selection of specific LWTs to go into further detail motivated by the big variability in the model ensemble for these LWTs. It would be interesting if the authors could comment on the importance of these specific LWTs (A-type, C-type, W-type and S-type) in the metrics you chose to define multi-hazard favouring conditions, i.e. F-Score and UHI metrics. How do the models perform in the LWTs that dominate those two metrics? Do the same models perform best/worst in both metrics? If not, which is the component that determines their performance for the two metrics, i.e. again which are the dominant weather types that determine the two indices? Can their assessment/results be used to recommend model(s) for specific hazard studies?

Also, the authors used two reanalysis datasets for the assessment of model performance. Looking at the LWTs presented in, e.g. Figure 2, the two reanalysis products differ quite significantly between each other and the observed Lamb Weather Types. Particularly in the F-score the 20th Century Reanalysis seems to overestimate the hazard quite significantly compared to observations and the NCEP reanalysis product. Can the authors link this difference in performance to specific characteristics of the two reanalysis products? Which one of the three metrics should the reader focus on when trying to assess model performance, i.e. is one of the datasets more/less reliable than the other(s)?  A comment on this would help the reader to draw a conclusion in regard to model performance. A dashed line in Figures 3, 5 and 6 indicating the value of observations/reanalysis would also help to clearly present model performance.

Discussion and Conclusions

Overall this section should link the statements made here to the main findings from the results section and highlight the most important results. Not every reader will go through the full results section. The authors should therefore repeat what their most important results are and which of their results they are referring to before commenting on implications of the change in specific variables or drawing comparisons to other studies. For instance, in line 444 the authors could include: “Greater A-Type persistence and frequency as found in our analysis…”.

Also, when highlighting differences between their results and previous studies the authors should indicate differences in the applied methods in their and the previous studies to help the reader draw conclusions about which new insights the authors provide in this manuscript and understand why there are differences in the results.

Detailed comments

Line 52 : Add Khanal et al. (2019) to the references for studies assessing river and coastal flooding

Khanal, S., Ridder, N., Terink, W. and Hurk, B.V.D., 2019. Storm surge and extreme river discharge: a compound event analysis using ensemble impact modelling. Frontiers in Earth Science7, p.224.

Line 62: Add Ridder et al. (2018) to reference of studies that investigated linkages between weather patterns/large-scale atmospheric circulation and local extreme events

Ridder, N., De Vries, H. and Drijfhout, S., 2018. The role of atmospheric rivers in compound events consisting of heavy precipitation and high storm surges along the Dutch coast. Natural Hazards and Earth System Sciences18(12), pp.3311-3326.

Lines 169, 451 and 456: Change “on the other hand” to “in contrast” (or similar) or add “on the one hand” to the statement that you are contrasting to

Line 453: Comment on the likelihood that your finding is an artefact of the applied methodology? And if it is an artefact, what does that mean for your conclusions?

Lines 479 – 483: I suggest moving (or repeating) this whole section “Recent analysis show that … and in improving resilience” to the introduction where there is no reference made to the link between LWTs and hazardous weather conditions. This would strengthen the motivation of the study and the choice of methodology.

Reviewer 3 Report

The project team has produced a very nice manuscript that is well organized and reads easily. The research is important for understanding future weather hazards given likely climate scenarios. My primary criticism of the manuscript is the limited historical dataset used, but it is limited by data availability. Well researched overall.
